# Photophysical image analysis for sCMOS cameras: Noise modelling and estimation of background parameters in fluorescence-microscopy images

**Dibyajyoti Mohanta**[1,2]*, **Radhika Nambannor Kunnath**[3]☯, **Erik Clarkson**[1]☯, **Albertas Dvirnas**[1,3], **Fredrik Westerlund**[3], **Tobias Ambjörnsson**[1]

**1** Centre for Environmental and Climate Science, Lund University, Lund, Sweden, **2** Department of Chemistry, The State University of New York at Buffalo, New York, United States of America, **3** Department of Life Sciences, Chalmers University of Technology, Gothenburg, Sweden

☯ These authors contributed equally to this work.
* dibyajyoti.mohanta@cec.lu.se

**Data availability statement:** The experimental data presented herein and MATLAB

## Abstract

Fluorescence microscopy is an effective tool for imaging biological samples, yet captured images often contain noises, including photon shot noise and camera read noise. To analyze biological samples accurately, separating background pixels from signal pixels is crucial. This would ideally be guided by the knowledge of a parameter called the Poisson parameter, $\lambda_{bg}$, representing the mean number of photons collected in a background pixel (for the case when quantum efficiency = 1 and the dark current is negligible).

This study introduces a method for estimating $\lambda_{bg}$, from an image which contains both background and signal pixels, using probabilistic noise modeling for an sCMOS camera. The approach incorporates Poisson-distributed photon shot noise and sCMOS camera read noise modelled with a Tukey-Lambda distribution. We apply a chi-square test and a truncated fit technique to estimate $\lambda_{bg}$ directly from a general sCMOS image, with camera parameters determined through calibration experiments.

We validate our method by comparing $\lambda_{bg}$ estimates in images captured by sCMOS and EMCCD cameras for the same field of view. Our analysis shows strong agreement for low to moderate exposure images, where estimated values for $\lambda_{bg}$ align well between the sCMOS and EMCCD images. Based on our estimated $\lambda_{bg}$, we perform image thresholding and segmentation using our previously introduced procedure.

Our publicly available software provides a platform for photophysical image analysis for sCMOS camera systems.

## Introduction

Fluorescence imaging of biological samples is fundamental in studying cellular and molecular components. Fundamentally, a fluorescence microscope operates on the principle of distributing a limited photon budget across the spatial dimensions of a detector or an image plane. Fluctuations due to a limited number of photons is referred to as photon shot noise [1].

implementations of the sCMOS-PMF pipeline are made available at (https://github.com/dibyajyoti41).

**Funding:** T.A. acknowledges a research grant 2022-03475 from the Swedish Research Council. F.W. and T.A. are grateful for funding from the Erling-Persson Foundation and the Swedish Childhood Cancer Fund (Barncancerfonden), Grant No. MT2022-003. The computations were enabled by resources provided by the National Academic Infrastructure for Supercomputing in Sweden (NAISS) and the Swedish National Infrastructure for Computing (SNIC) at the national Supercomputing Centre at Linköping University partially funded by the Swedish Research Council through grant agreements no. No. 2022-06725 and No. 2018-05973. The funders had no role in study design, data collection and analysis, decision to publish, or preparation of the manuscript.

**Competing interests:** The authors have declared that no competing interests exist.

In addition, the image recorded in fluorescence microscopy is affected by the camera read noise which is the random electronic noise introduced by the sensor's readout electronics [2,3].

Scientific complementary metal oxide semiconductor (sCMOS) cameras have gained significant popularity in recent years for imaging dynamic biological samples [4,5]. Their advantages, such as higher frame rates (images captured per second), larger sensor areas, high quantum efficiency (photon-to-photoelectron conversion rate), and low effective read noise , make them an increasingly popular alternative to EMCCD cameras, which are limited by multiplicative noise (noise multiplied along with the signal by the electron multiplying gain) [6–9]. Processing sCMOS fluorescence images (i.e. image thresholding, image segmentation) still poses challenges, as each pixel may exhibit independent response characteristics, such as noise and offset, due to the independent readout units [2,7]. This is also different in comparison to EMCCD where the incoming photoelectrons are shifted serially through a gain multiplication register [10].

A common challenge in fluorescence microscopy image analysis is image thresholding, which distinguishes signal pixels (emitted photons from fluorophores) from background pixels (dominated by non-specific photons). Thresholding methods are categorized into supervised and unsupervised approaches. Supervised methods require manual parameter adjustments tailored to each image or extensive labeled training data, while unsupervised techniques automatically classify pixels based on heuristic criteria or intrinsic image properties such as photon statistics and intensity distributions or calibration data [10]. In low-intensity images, distinguishing signal from background becomes especially difficult due to minimal contrast and overlapping photon distributions [9]. However, incoming photons follow probability distributions, and understanding these statistical patterns can allow for improved image classification. It is the overarching purpose of this study to introduce a new probabilistic photophysics-based method for unsupervised image thresholding for low-intensity sCMOS images.

The gold standard for unsupervised image thresholding is the Otsu method, which has been widely used for decades. Otsu's approach is a heuristic method that finds an optimal intensity threshold by minimizing the weighted sum of intra-class variances (or equivalently maximizes inter-class variances) [11]. The Otsu method does not explicitly account for the noise characteristics inherent in imaging systems, nor does it provide rate of misclassifications - issues that are especially important in low-intensity image analysis.

Probabilistic unsupervised image thresholding methods include the Poisson-Gaussian (PG) framework which seeks to separate photon shot noise (Poisson) and stationary noise sources (Gaussian) [12], Generalized Anscombe Transformation (GAT)-based denoising [13], and pixel-wise MLE calibration [14]. However, these methods oversimplify noise sources by modeling them (including read noise) collectively as Gaussian with pixel-dependent variances. Recently, K. Wei et al [15], addressed the non-Gaussian nature of sCMOS read-noise featuring heavy-tail behavior in low-light regime images. They subsequently used a convolutional neural network (CNN) based image enhancing algorithm, which required large training datasets.

Despite advances in camera-specific noise modeling, a universal physics-driven framework for unsupervised probabilistic thresholding, with *a priori* control over misclassification rates, is lacking. To remedy this, we adapt the EMCCD photophysical image analysis pipeline by J. Krog et al [10] to sCMOS images. Following [15], we model the sCMOS readout noise using a Tukey-Lambda distribution to deal with low-light conditions. Our framework automatically estimates the background Poisson parameter ($\lambda_{bg}$) without user intervention directly from an image that contains both background and signal pixels (at arbitrary ratios). Using this

estimate, we subsequently perform unsupervised probabilistic sCMOS image thresholding and segmentation.

The paper is structured as follows: In "Materials and methods" we outline the method, including noise modeling at different sCMOS imaging stages, the probabilistic model for intensity distribution, and techniques for estimating camera parameters. In "Results" we present results, detailing noise model parameter estimation, Poisson parameter $\lambda_{\mathrm{bg}}$ for background pixels, and probabilistic image thresholding and segmentation. We also demonstrate the robustness of our approach by comparing estimates of $\lambda_{\mathrm{bg}}$ in images acquired from both sCMOS and EMCCD cameras of the same field-of-view (FOV) with identical experimental settings. In the "Summary and outlook" section, we conclude with key findings and future directions.

## Materials and methods

The photophysical image analysis pipeline integrates three components. First, modeling the sCMOS imaging cascade: from photon-to-photoelectron conversion, charge transfer, and amplification to digitization, with stage-specific noise sources (shot noise, read noise, quantization noise), see subsection "Theory" below. The model is expressed as an expression for the probability mass function (PMF) of the recorded image counts. Second, empirically estimating sCMOS camera parameters (gain, offset and the Tukey-Lambda distribution parameters) to align model with experimental data, see subsections "Experiments" and "Camera parameter estimation". Third, equipped with camera parameters, a probabilistic method estimating $\lambda_{bg}$ and performing photophysical image thresholding and segmentation, see the last subsections "Estimating $\lambda_{bg}$" and "Unsupervised probabilistic image thresholding and image segmentation".

### Experiments

Two types of cameras were used in this study: 1) sCMOS (Photometrics Prime 95B 22 mm) and 2) EMCCD (Andor iXon Ultra). The sCMOS camera has a pixel size of 11 $\mu$m and a sensor format of 1412 x 1412 pixels. The camera has three software-controllable gain modes: sensitive, balanced, and full well [16]. Throughout this study, we consistently used the balanced gain setting mode for all sCMOS camera experiments. The EMCCD camera has a pixel size of 13 $\mu$m and a sensor format of 1024 x 1024 pixels [16]. The EM gain setting of the camera is variable from 2 to 300.

All the images were captured using a Zeiss Axio Observer Z1 microscope which has two side ports to mount the cameras and a software option to switch between the camera ports. The microscope is coupled to a Colibri 5 multicolor LED light source that can emit four wavelengths. A 100X oil immersion microscope objective lens was used to capture all the signal images at a scaling of 110 nm per pixel.

In the following, we categorize the experiments performed in this study in three subsections i) Calibration experiments ii) fluorescent DNA and bacterial cell imaging experiments iii) Dual camera same FOV experiments.

**sCMOS calibration experiments.** We performed two types of sCMOS calibration experiments:

In the *Cap-on experiment*, the illumination was turned off and the camera shutter was kept closed to capture a dark-field image of the sCMOS camera with exposure time 1 $\mu$s. We estimated offset and read noise parameters by analyzing this image (see camera parameter estimation).

In the *Illuminated white wall experiment*, bright field movies were captured by focusing the sCMOS camera on a uniformly illuminated white wall and varying the illumination intensity from 10% to 100% in increments of 10 percentage points. We captured 1000 time frames at each intensity keeping the exposure time fixed at 400 ms. We used these stacks of bright images of different intensities to determine the gain by a mean-variance analysis 19.

**sCMOS images of fluorescent DNA and bacterial cells.** Fluorescently stained DNA was stretched on a silanized glass slide [17]. The DNA was stained with YOYO-1 dye that has an absorption maximum of 489 nm and an emission maximum of 509 nm. In the *Fluorescent DNA experiments*, these DNA molecules were imaged with a sCMOS camera with blue light excitation (469/38 nm bandpass excitation filter) at 8% illumination intensity and exposure time varying from 1 ms to 1000 ms [18].

For the cellular imaging experiments purpose, *Bacillus subtilis* bSS82 cells (genotype trpC2 amyE::spc PrpsD-gfp) which overexpress the green fluorescent protein (GFP) were used [19]. Live bacterial cells were fixed on agarose pads on microscopy glass slides. During imaging, the light intensity was fixed at 8% and the images were captured using the sCMOS camera at the following exposures: 10, 100, and 400 ms.

**Dual camera imaging of the same FOV.** We mounted the sCMOS and the EMCCD cameras on each of the two mounting ports (left and right) of the microscope. Using the fluorescently stained DNA sample on glass (prepared as described above), we captured an image using one camera, and then switched port and captured the same field of view (FOV) using the other camera. The camera port was switched using the microscope software. We recorded the images in both cameras under identical experimental conditions (same light source intensity, and exposure time). In these experiments, the EMCCD camera had a gain setting of 100 and the sCMOS camera used the balanced-gain setting. To minimize the effects of photobleaching, a different FOV on the glass slide was imaged for each exposure setting and the videos were limited to 20 frames. Also, the cameras were alternated, in the sense that if an image was captured at one exposure setting first with sCMOS and then EMCCD, the image for the next exposure setting was captured first with EMCCD and then sCMOS.

In addition, to estimate the camera model parameters of the EMCCD camera (required for analysis of the same FOV images of the EMCCD), we performed a set of calibration experiments (*Cap-on* and *Illuminated white wall*) similar to the sCMOS camera calibration [10]. To this end, the dark field image was captured at a minimum EM gain 2 and at a fixed exposure of $1\mu$s. We recorded 100 bright field time frames by EMCCD camera at each different intensity (10%, 20%, $\cdots$, 100%) keeping exposure time and EM gain fixed at 500 ms and 100, respectively.

## Theory

The output from each image pixel is a digital image count ($n_{ic}$), which can be expressed as a sum of independent random contributions,

$$n_{ic} = g\,n_{oe} + N_{\text{read}} + N_q + \Delta, \tag{1}$$

where $n_{oe}$ is the number of output photoelectrons generated in each pixel area. $N_{read}$ represents the cumulative noise generated in the readout circuit during the conversion of photoelectrons to voltage signals. $g$ is a constant representing overall gain of the system in units of ADU/$e^-$, during the conversion of electrons to digital counts. $N_q$ represents the quantization error (in digital counts or ADU) during conversion to a digital number, and $\Delta$ is the offset, a constant (in ADU) added to each pixel to prevent any undesirable negative output.

The quantities $n_{oe}$ due to the quantum nature of light, $N_{\text{read}}$ due to electronic fluctuations in the read-out circuitry, and $N_q$ due to rounding to the next integer value, are random numbers, and as a consequence, $n_{ic}$ is also a random variable. The aim of the modelling is to derive an expression for the probability mass function (PMF) of $n_{ic}$. To this end, we start by briefly discussing the generation of each of these components and the underlying processes, before deriving an explicit formula for the PMF.

**Photons hit the sensor region.** In an imaging process, photons arriving at the camera sensor are typically assumed to be Poisson-distributed [10,15]. This underpins the principal noise source in sCMOS imaging: photon shot noise, which is non-deterministic due to the quantum nature of light, and varies randomly about a mean number of photons ($\Lambda$) directly proportional to the camera's exposure time. These photons then undergo photoelectric conversion to the mean number of photoelectrons ($\lambda$) at the sensor. The conversion factor is known as quantum efficiency (QE). The resulting random variable $n_{oe}$ has a PMF:

$$p(n_{oe}|\lambda) = \mathcal{P}(n_{oe}; \lambda), \quad \lambda = \text{QE} \cdot \Lambda + c \tag{2}$$

where $\mathcal{P}(k; \lambda)$ is the PMF for the Poisson distribution with parameter $\lambda$. $c$ is a constant corresponding to the mean number of photoelectrons in the absence of incoming photons (e.g. dark currents) [20].

**Transfer of photoelectrons through readout circuitry.** After the photoelectric conversion of photons to electrons within each pixel area of the sensor region, these electrons are amplified and converted into voltage signals by the output readout circuitry inside each pixel [2,16]. The conversion of electrons ($e^-$) to voltage signals at each pixel is mediated by a multiplication factor called the analog conversion gain ($g'$). Usually, the voltage signals and $g'$ are expressed in terms of $\mu$V and $\mu$V/$e^-$, respectively.

The readout process in sCMOS sensors introduces read noise ($N_{\text{read}}$), which arises from electronic perturbations during charge-to-voltage conversion in the pixel's readout circuitry. Such noise also includes contributions from thermal noise and source follower noise [15]. Under certain conditions (e.g., high readout speeds), source follower noise can exhibit non-Gaussian outliers, leading to a long-tailed distribution that deviates from purely Gaussian behavior [15]. Recent work argues that the Tukey-Lambda (TL) distribution family can effectively model the long-tail behavior of read noise ($N_{\text{read}} \sim \text{TL}(\Lambda_{\text{TL}}, \mu_{\text{TL}}, \sigma_{\text{TL}})$ in sCMOS cameras. The TL distribution is defined by its quantile function for a uniformly distributed random variable $R \sim \mathcal{U}(0, 1)$ [21]:

$$N_{\text{read}} = \mu_{\text{TL}} + \sigma_{\text{TL}} \, Q(R; \Lambda_{\text{TL}}), \tag{3}$$

with

$$Q(R; \Lambda_{\text{TL}}) = \begin{cases} \dfrac{R^{\Lambda_{\text{TL}}} - (1-R)^{\Lambda_{\text{TL}}}}{\Lambda_{\text{TL}}}, & \text{if } \Lambda_{\text{TL}} \neq 0, \\ \ln\left(\dfrac{R}{1-R}\right), & \text{if } \Lambda_{\text{TL}} = 0, \end{cases} \tag{4}$$

where $\Lambda_{\text{TL}}$ is the shape parameter that determines the type of distribution, such as $\Lambda_{\text{TL}} \approx 0.14$ approximates a Gaussian distribution and $\Lambda_{\text{TL}} = 0$ corresponds to a logistic distribution. $\mu_{TL}$, the location parameter is set to zero following the zero mean noise assumption [15]. $\sigma_{TL}$ is the scale parameter.

Each pixel in a row is then connected to the appropriate column voltage bus, where the on-chip analog-to-digital conversion (ADC) process in sCMOS cameras maps a pixel's voltage

signal $V_{\text{pixel}}$ (voltage signal converted from the charge of electrons in a given pixel) to a digital value (ADU) using the relation: $V_{\text{pixel}} \times \frac{(2^{n_b}-1)}{V_{Ref}}$, where $n_b$ is the bit depth (resolution) of the ADC, and $V_{\text{Ref}}$ is the ADC's reference voltage (full-scale range) [16,22]. Therefore, the overall gain $g$ of the sCMOS camera is equal to the magnitude of analog gain $g'$ (charge-to-voltage conversion factor) with units $[\mu V/e^-] \times [ADU/\mu V] = ADU/e^-$, where $ADU/\mu V$ represents the unit of transfer function of ADCs [23]. We operated in balanced mode with 12-bit depth ADC, which balances sensitivity and noise [16,24].

The conversion of overall voltage signal ($V_{pixel}$) within a pixel to the nearest integer introduces a quantization noise (in units of ADU) which approximately follows a uniform distribution [10],

$$N_q \sim U\left(\frac{-1}{2q}, \frac{1}{2q}\right) \tag{5}$$

with $q = 1$ is the quantization step [15].

After the conversion processes described above, we have $g\,n_{oe} + N_{read} + N_q$ for each pixel in units of ADU. To this number, the camera adds offset ($\Delta$) to form the final output image count ($n_{ic}$) given by Eq (1).

## PMF for the image counts

In the above section, we have seen that the random variables $n_{oe}$, $N_{read}$ and $N_q$ have distinct sources of origin and are characterized by PDFs/PMFs of known form. In such cases, the characteristic function (CF), which is the Fourier transform of the PMF given by $\langle \exp(ikn_{ic}) \rangle$, is a useful construct. The CF of a sum of independent random variables factorizes into the CFs of the individual random variables [25]. Hence, we may first calculate the CF and then Fourier-invert it back to its corresponding PMF. So, the characteristic function of the recorded image count $n_{ic}$ (see Eq (1)) in a pixel is given by,

$$\phi(k) = \langle \exp(ikn_{ic}) \rangle = \underbrace{\langle \exp(ikg n_{oe}) \rangle}_{\text{Photoelectrons}} \underbrace{\langle \exp(ikN_{\text{read}}) \rangle}_{\text{Read Noise}} \underbrace{\langle \exp(ikN_q) \rangle}_{\text{Quantization Noise}} \exp(ik\Delta) \tag{6}$$

where $k$ is the Fourier variable associated with $n_{\text{ic}}$. As the mean number of incoming photons and outgoing photoelectrons ($\Lambda$ and $\lambda$, respectively) both follow Poisson distributions, the individual characteristic function of the outgoing photoelectrons $\lambda$ has a closed form [26] and is represented by,

$$\langle \exp(ikg\,n_{oe}) \rangle = \exp\left(\lambda\left(e^{ikg} - 1\right)\right) \tag{7}$$

with $\lambda = \text{QE} \cdot \Lambda + c$ as before.

The CF of the read-noise $N_{\text{read}}$ (3), is:

$$\langle \exp(ikN_{\text{read}}) \rangle = \int_0^1 \exp\left(ik\sigma_{\text{TL}} \frac{R^{\Lambda_{\text{TL}}} - (1 - R)^{\Lambda_{\text{TL}}}}{\Lambda_{\text{TL}}}\right) dR. \tag{8}$$

where the choice of integration limits follows from the fact that expectation value on the right-hand side is with respect to a uniform distribution over the range [0,1]. The integral above cannot be solved analytically. Thus, we approximate it by discretizing $R$ into $n$ points: $R_l = (l - 1)\delta$, $\delta = 1/(n - 1)$, $l = 1, \ldots, n$. We evaluate the integrand at each $R_l$:

$$y_l = \exp\left(ik\sigma_{\text{TL}} \frac{R_l^{\Lambda_{\text{TL}}} - (1 - R_l)^{\Lambda_{\text{TL}}}}{\Lambda_{\text{TL}}}\right). \tag{9}$$

and, lastly we apply the trapezoidal rule:

$$\langle \exp(ikN_{\text{read}}) \rangle \approx \frac{\delta}{2} \left( y_1 + y_n + 2 \sum_{l=2}^{n-1} y_l \right). \tag{10}$$

Next, the CF of the quantization noise $N_q$ (Eq 5) is given by:

$$\langle \exp(ikN_{q=1}) \rangle = \frac{\sin(k/2)}{k/2}. \tag{11}$$

Combining all independent components, the total CF becomes:

$$\Phi(k) \approx \exp\left( \lambda \left( e^{ikg} - 1 \right) \right) \left[ \frac{\delta}{2} \left( y_1 + y_n + 2 \sum_{l=2}^{n-1} y_l \right) \right] \frac{\sin(k/2)}{k/2} \exp(ik\Delta) \tag{12}$$

where $y_l$ is given in equation (9).

To compute the PMF $p(n_{ic})$, we apply the Gil-Pelaez Fourier inversion [27] to the CF $\Phi(k)$:

$$p(n_{ic}) = \text{PMF}(n_{ic}) \approx \frac{1}{\pi} \int_0^\pi \mathcal{R} \left[ \Phi(k) e^{-ikn_{ic}} \right] dk, \tag{13}$$

where $\mathcal{R}$ denotes the real part of the integrand. The upper limit is $\pi$ because the CF of the discrete variable $n_{ic}$ is periodic with period $2\pi$, so the inversion integral is taken over $[0, \pi]$. We again discretize the integral by using a trapezoidal quadrature similar to Eq (10).

The cumulative distribution function $\text{CDF}(n_{ic})$ is calculated by the summation of PMF $p(n_{ic})$ [10]

$$\text{CDF}(n_{ic}) = \sum_{n=0}^{n_{ic}} p(n). \tag{14}$$

We use this CDF for the image thresholding algorithm which discriminates background from signal pixels based on their probabilistic distributions.

The sCMOS-PMF given by the Eq (13) is different from EMCCD-PMF [10] in two fundamental ways. First, here the gain step does not include electron-multiplication (the sCMOS camera does not amplify the converted output electrons). Second, the read noise for sCMOS cameras is a Tukey-Lambda distributed random number instead of a Gaussian random number as for EMCCD [10].

## Camera parameter estimation

We here demonstrate how to estimate the camera model parameters, *chipParams* = $(g, \Lambda_{\text{TL}}, \sigma_{\text{TL}}, \Delta)$, through theoretical analysis of calibration experiments. In the "Results" section, we will use *chipParams* in the estimation of $\lambda_{bg}$.

**Estimation of the gain parameter, *g*.** We estimate the sCMOS camera gain, $g$, using a mean-variance approach, where $g$ is obtained as the slope of the experimental pixel-based mean-variance relationship.

For the mean-variance analysis we use data from the *Illuminated white wall experiment* (see subsection "Experiments"), at all different illumination intensities. For each pixel, the experiments provide a time series of the image counts from which we estimate the means and the variances. This procedure yields experimental mean-variance data, i.e., $(\overline{n}_{ic}^{(j)}, S^{2(j)})$, where $j$ labels the pixels in all experiments ($j = 1, \dots, m$, where $m$ is the number of pixels).

To match the experimental mean-variance data to theory, we need to derive an expression for the relation between the mean image count and its variance. Taking the expectation value of Eq (1), we obtain the mean image count at a pixel as

$$\mathrm{E}[n_{ic}] = g\,\mathrm{E}[n_{oe}] + \mathrm{E}[N_{read}] + \mathrm{E}[N_q] + \Delta. \tag{15}$$

Since $n_{ic}$ is a sum of independent random variables (discussed above), its mean is simply the sum of the individual means. Furthermore, using the facts that $\mathrm{E}[n_{oe}] = \lambda$, $\mathrm{E}[N_{read}] = 0$ (since $\mu_{TL} = 0$) and $\mathrm{E}[N_q] = 0$ we obtain

$$\mathrm{E}[n_{ic}] = g\lambda + \Delta. \tag{16}$$

Similarly, the variance of $n_{ic}$, see Eq (1), is obtained using the fact that variances of independent random variables add up:

$$\mathrm{Var}[n_{ic}] = g^2\lambda + s_{\mathrm{TL}}^2 + \frac{1}{12}. \tag{17}$$

where we used that for a Poisson distribution, the mean is equal to the variance, i.e. $\mathrm{E}[n_{0e}] = \mathrm{Var}[n_{oe}] = \lambda$. The variance of Tukey-lambda distributed random number is given by,

$$s_{\mathrm{TL}}^2 = \sigma_{\mathrm{TL}}^2 \left[ \frac{2}{\Lambda_{\mathrm{TL}}^2} \left( \frac{1}{1 + 2\Lambda_{\mathrm{TL}}} - \frac{\Gamma(1 + \Lambda_{\mathrm{TL}})^2}{\Gamma(2\Lambda_{\mathrm{TL}} + 2)} \right) \right], \tag{18}$$

where $\Lambda_{TL} > -1/2$ [21] and $\Gamma(z)$ is the Gamma function. Above we also used the fact that the variance of the rounding error $N_q$ is given by $\mathrm{Var}(N_{q=1}) = 1/12$ [10,28].

Solving Eq (16) for $\lambda$ and plugging the result into Eq (17) we obtain our final mean (subtracted by offset)-variance relation:

$$\mathrm{Var}[n_{ic}] = g\,(\mathrm{E}[n_{ic}] - \Delta) + d. \tag{19}$$

where the constant ("y-intercept")

$$d = s_{\mathrm{TL}}^2 + \frac{1}{12}. \tag{20}$$

Inspecting Eq (19), we notice that the gain, $g$, appears as the slope if the variance is plotted as a function of the offset subtracted from mean ($\mathrm{E}[n_{ic}] - \Delta$). In the mean-variance analysis, we therefore fit (using the least squares method) our experimental mean-variance data (see above) to a straight line. The slope of this line serves as our estimate of $g$. The fitted value for the constant $d$ is here not used for parameter estimation. We will, however, later show how to use it as a consistency check.

**Offset estimation.** The offset, $\Delta$, is estimated by using the dark frame image acquired during *Cap-on experiment* (see the "Experiments" section). In this experiment, we have no input photons and therefore we can set the $n_{oe}$ to 0 in Eq (1). By further taking the expectation value of this equation we obtain $E[n_{ic}] = \Delta$ following Eq (15). Hence, $\Delta$ is the average image count in the *Cap-on experiment*. We therefore estimate $\Delta$ as the empirical mean,

$$\Delta = \frac{1}{m} \sum_{j=1}^{m} n_{ic}^{(j)} \tag{21}$$

where $n_{ic}^{(j)}$ is the recorded image count for pixel $j$ ($j = 1, 2, .., m$) in the cap-on experiment.

**Estimation of the read noise parameters ($\Lambda_{TL}$, $\sigma_{TL}$).** To estimate the shape parameter ($\Lambda_{TL}$) and the scale parameter ($\sigma_{TL}$) in the Tukey-Lambda distribution we again make use of the *Cap-on experiment*. Again setting the first term in Eq (1) to 0 (no incoming photons in the cap-on experiments), we find that the image count in these experiments is a random variable $n_{ic} = N_{\mathrm{read}} + N_q + \Delta$. In the following, we also neglect the rounding error, i.e., we set $N_q \approx 0$, and assume $\Delta$ known (i.e. estimated in the previous subsubsection). With these approximations, we have $n_{ic} - \Delta \approx N_{\mathrm{read}}$, i.e., the statistics of the recorded image count in the *Cap-on experiment* (with the offset subtracted) is described by the Tukey-Lambda distribution.

To estimate the parameters $\Lambda_{TL}$ and $\sigma_{TL}$, we use a technique called probability plot correlation coefficient (PPCC) [21,29], which we here recapitulate for completeness. The PPCC method, as applied to our data, is divided into the following steps:

- Collect the image counts from a cap-on sCMOS image into an one-dimensional array. Sort this array to yield: $(n_{ic}^{(1)}, n_{ic}^{(2)}, \ldots, n_{ic}^{(m)})$, where $n_{ic}^{(1)} \leq n_{ic}^{(2)} \leq \ldots \leq n_{ic}^{(m)}$.
- Write the theoretical quantiles for the Tukey-Lambda distribution on the form

$$Q_{\mathrm{theory},j} = \sigma_{\mathrm{TL}} F_{\mathrm{theory},j} \tag{22}$$

for $j = 1, \ldots, m$, with unscaled theoretical quantiles

$$F_{\mathrm{theory},j} = \frac{u_j^{\Lambda_{\mathrm{TL}}} - (1 - u_j)^{\Lambda_{\mathrm{TL}}}}{\Lambda_{\mathrm{TL}}}, \tag{23}$$

where

$$u_j = \frac{j - 0.5}{m}.$$

For a fixed set of parameters, the quantile function for the Tukey-Lambda distribution can be graphically illustrated by plotting $\mathbf{Q}_{\mathrm{theory}}$ as a function of $\mathbf{u}$ where $\mathbf{Q}_{\mathrm{theory}} = (Q_{\mathrm{theory},1}, Q_{\mathrm{theory},2}, \ldots, Q_{\mathrm{theory},m})$ and $\mathbf{u} = (u_1, u_2, \ldots, u_m)$ (by instead plotting $\mathbf{u}$ as a function of $\mathbf{Q}_{\mathrm{theory}}$, one could illustrate the cumulative distribution function).
- Compute empirical quantiles

$$Q_{\mathrm{emp},j} = n_{ic}^{(j)} - \Delta, \quad j = 1, \ldots, m.$$

- For each $\Lambda_{TL}$, within a range $[-1,1]$, compute the Pearson correlation coefficient (PCC) between the $\mathbf{Q}_{\mathrm{emp}}$ and $\mathbf{F}_{\mathrm{theory}}$:

$$\rho(\Lambda_{TL}) = \frac{\sum_{j=1}^{m} (Q_{\mathrm{emp},j} - \bar{Q}_{\mathrm{emp}})(F_{\mathrm{theory},j} - \bar{F}_{\mathrm{theory}})}{\sqrt{\sum_{j=1}^{m} (Q_{\mathrm{emp},j} - \bar{Q}_{\mathrm{emp}})^2 \sum_{j=1}^{m} (F_{\mathrm{theory},j} - \bar{F}_{\mathrm{theory}})^2}}. \tag{24}$$

The PCC takes values between $-1$ and $1$, and perfect agreement (up to a scale factor) gives PCC = 1. For a "good fit" ($\mathbf{Q}_{\mathrm{theory}} \approx \mathbf{Q}_{\mathrm{emp}}$), a quantile-quantile (QQ) plot of $\mathbf{Q}_{\mathrm{emp}}$ versus $\mathbf{F}_{\mathrm{theory}}$ should ideally be a straight line, where the slope of this line gives $\sigma_{TL}$ (see Eq 22) [29]. We therefore choose the value of $\Lambda_{TL}$ that maximizes $\rho$ [29] (the "optimal" $\Lambda_{TL}$).
- For the optimal $\Lambda_{TL}$ from above, we then estimate the slope by fitting a linear function using the least squares method; the value of the fitted slope serves as our estimate of $\sigma_{TL}$ [29].

**Consistency check of estimated camera parameters.** After estimating all the camera parameters: $chipParams = (g, \Lambda_{\mathrm{TL}}, \sigma_{\mathrm{TL}}, \Delta)$, as a consistency check, we plug these estimates into Eqs (18) and (20) to obtain an expected value for the y-intercept, $d$. This value can then be compared to the actual y-intercept obtained in the fitting procedure in the mean-variance analysis in Fig 1.

## Estimating $\lambda_{bg}$

Using the estimated sCMOS camera chip parameters, $chipParams$, we estimate the Poisson parameter $\lambda_{\mathrm{bg}}$ describing background pixels in sCMOS images containing both background and signal regions. Following Krog et al. [10], we fit a truncated version of sCMOS-PMF, Eq (13) given by $\mathrm{PMF}(n_{ic}^{(j)}|\theta)$, to the lower tail of the image count histogram (where background dominates) with $\lambda_{bg}$ as a fit parameter. Note that we deliberately restrict the fit to the lower tail of the image count histogram, so that any contribution from signal photons (described by Poisson parameter $\lambda_{\mathrm{sig}}$) does not enter our truncated PMF fit. The fitting procedure involves the truncated likelihood: $\prod_j \frac{\mathrm{PMF}(n_{ic}^{(j)}|\theta)}{\mathrm{CDF}(N_{ic}^{\mathrm{bg}}|\theta)}$, where $\theta = \{chipParams, \lambda_{bg}\} = \{g, \Lambda_{\mathrm{TL}}, \sigma_{\mathrm{TL}}, \Delta, \lambda_{bg}\}$.

As in [10], we iteratively adjust the truncation point $N_{ic}^{\mathrm{bg}}$ and perform maximum likelihood estimation (MLE) until the fit satisfies a goodness-of-test with a significance level set by $p_{\mathrm{GoF}} = 0.01$.

## Unsupervised probabilistic image thresholding and image segmentation

With all camera parameters and $\lambda_{\mathrm{bg}}$ estimated, we can perform unsupervised probabilistic image thresholding and segmentation, similar to [10], but here for an sCMOS imaging system instead of an EMCCD setup.

For image thresholding, we follow the procedure formalized by [10]. We first estimate an image count threshold, $N_{ic}^{\mathrm{thresh}}$ based on a p-value binarization threshold $p_{\mathrm{binarize}} = 0.01$. Here,

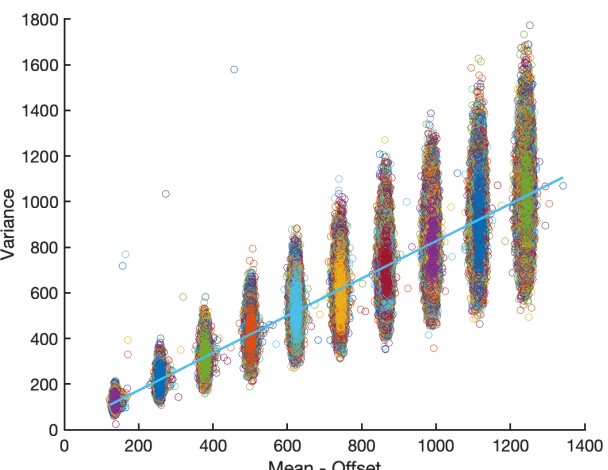

**Fig 1. Estimating the gain parameter, *g*, of the sCMOS camera: Variance vs mean plot of bright field images acquired during the *Illuminated white wall experiment* (see 'Experiments' section).** Intensity levels were incremented in 10% steps from 10% to 100%. The gain is calculated from the slope of plot using the Eq (19), yielding $g = 0.81 \pm 0.005$ (shown as the solid cyan line). The y-intercept of the linear fit is 6.4. This value is consistent with the value 4.7 obtained using Eq (20).

$p_{\text{binarize}}$ is our a priori value of the number of false positives that we accept (i.e., an acceptable value for the fraction of white pixels in background regions). The p-value is turned into an image count threshold, $N_{\text{ic}}^{\text{thresh}}$ by inverting the CDF (Eq 14). With this threshold in hand, we binarize (threshold) the image by turning pixels with an image count above the threshold, white, and those below the threshold, black. The strength of this method is that by using $p_{\text{binarize}}$ we know a priori the expected fraction of white pixels in background regions.

For segmentation, we use the binarized image and apply the method from [10] with allowedGapLength = 1 to identify the connected components of white/black pixels. Segmentation quality is controlled via a p-value $p_{\text{seg}}$ (see [10] for details), calculated using the sCMOS-PMF for summed counts in each segmented region.

## Results

The Results section is organized as follows: First, we calibrate the sCMOS parameters (gain, offset, read noise parameters) using calibration experiments. Next, we estimate the background Poisson parameter ($\lambda_{bg}$) from images containing both signal and background pixels. We then perform automated, unsupervised thresholding and segmentation with prior error estimation, requiring no user intervention. Finally, we validate the sCMOS-PMF framework by comparing $\lambda_{bg}$ estimates derived from sCMOS and EMCCD detectors under identical imaging conditions, confirming the accuracy of our algorithm.

### Camera parameter estimation

Using a set of calibration experiments combined with the parameter estimation procedure described in Materials and Methods, we estimate the model parameters for the sCMOS camera used herein.

We first estimate gain parameter of the camera using data from the *Illuminated white wall experiment* recorded at increasing intensities (10%,20%,..,100%), see Fig 1. Through mean-variance analysis (see "Methods and materials"), the slope of this relationship yields the camera's gain, estimated as $g = 0.81 \pm 0.005$ ADU/$e^-$ under balanced conversion gain settings.

From the mean dark-frame intensity in the *Cap-On Experiment* (Eq 21), we estimate $\Delta = 100.10 \pm 0.05$ ADU, which is in close agreement to the factory default bias of 100 ADU [16].

We next seek to estimate the camera parameters associated with the read noise. From our *Cap-On experiment*, we acquired the dark frame images, which are used to estimate shape parameter ($\Lambda_{TL}$) and scale parameter ($\sigma_{TL}$) in the Tukey-Lambda distribution family following the steps discussed in "Methods and materials" (see Fig 2). The mean value of $\Lambda_{TL}$ and $\sigma_{TL}$ over all dark frames of our sCMOS camera are given by values $0.055 \pm 0.007$ and $1.310 \pm 0.005\, e^-$, respectively.

### Estimating $\lambda_{bg}$ in an image with background and signal pixels

Using the calibrated camera parameters ($g$, $\Lambda_{\text{TL}}$, $\sigma_{\text{TL}}$, $\Delta$), we can estimate the Poisson parameter $\lambda_{\text{bg}}$ describing background pixels in sCMOS images containing both background and signal regions using the procedure described in Materials and Methods. We estimate the background Poisson parameter $\lambda_{\text{bg}}$ of a fluorescence image of DNA on glass (Fig 3), acquired with a 100 ms exposure time and balanced gain settings.

To deal with non-uniform illumination, like in [10], we partition the image into $16 \times 16$ tiles ($64 \times 64$ pixels each). Focusing on the tile {(7,13)}, we compute $\lambda_{\text{bg}}$ using our truncated PMF fitting procedure described in Materials and Methods. In Fig 3(b), the histogram's blue

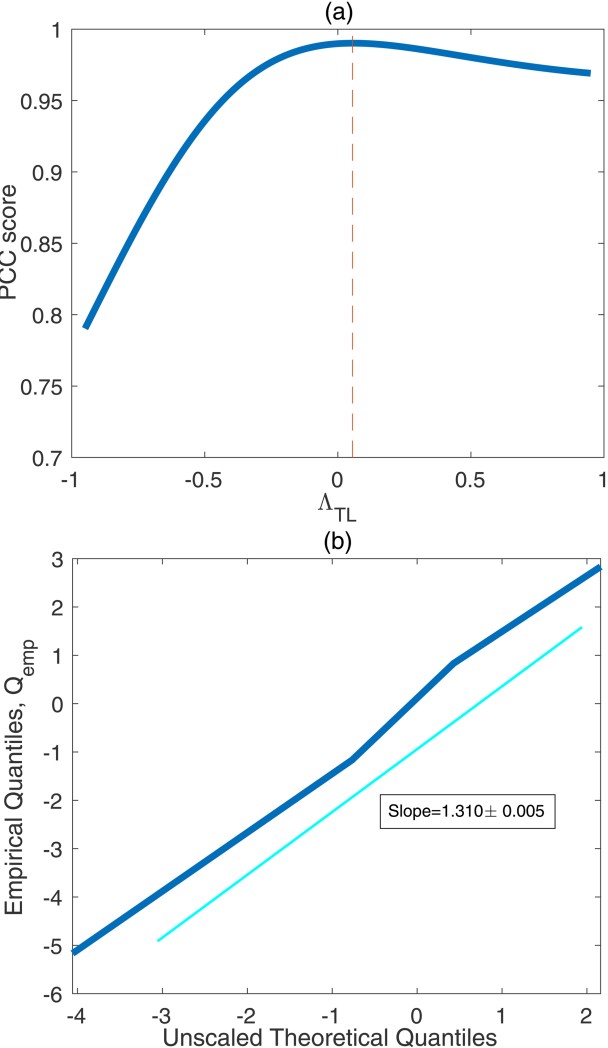

**Fig 2. Estimation of read noise parameters, $\Lambda_{TL}$ and $\sigma_{TL}$, for the sCMOS camera.** Panel (a) shows the Pearson correlation coefficient, Eq (24), between the empirical quantiles , $\mathbf{Q}_{emp}$, from the dark frame image (acquired during *Cap-On experiment*) and the unscaled theoretical quantiles, $\mathbf{F}$, for a range of shape parameters ($\Lambda_{TL}$). The dashed line in the figure represents $\Lambda_{TL} = 0.055 \pm 0.007$ corresponding to the maximum PCC score. Panel (b) plots the dark image frame empirical quantiles against the theoretical unscaled quantiles at the optimal $\Lambda_{TL}$ from panel (a). The slope of this curve gives estimation of the scale parameter ($\sigma_{TL}$) given by $1.310 \pm 0.005 \, e^-$ . The y-intercept of the fitted line is 0.009 (which is close to the expected value 0).

bars represent pixel intensities below $N_{ic}^{bg}$ = 190, identified as true background, while orange bars denote uncertain (mixed background/signal) pixels. Our analysis yields $\lambda_{bg}$ = 102.6, corresponding to an average of approximately 102 photoelectrons (or 102 photons multiplied with QE) generated in the sensor. To illustrate the robustness of our algorithm, we also fit the image counts from the tile {(8,8)} in Fig 3(c). Here, the estimated $\lambda_{bg}$ = 102.2 remains similar to that of {(7,13)}. However, the threshold $N_{ic}^{bg}$ increases to 197 highlighting the variation in the image. The sCMOS-PMF given by black dashed line, Eq (13) fits well to the entire image histogram.

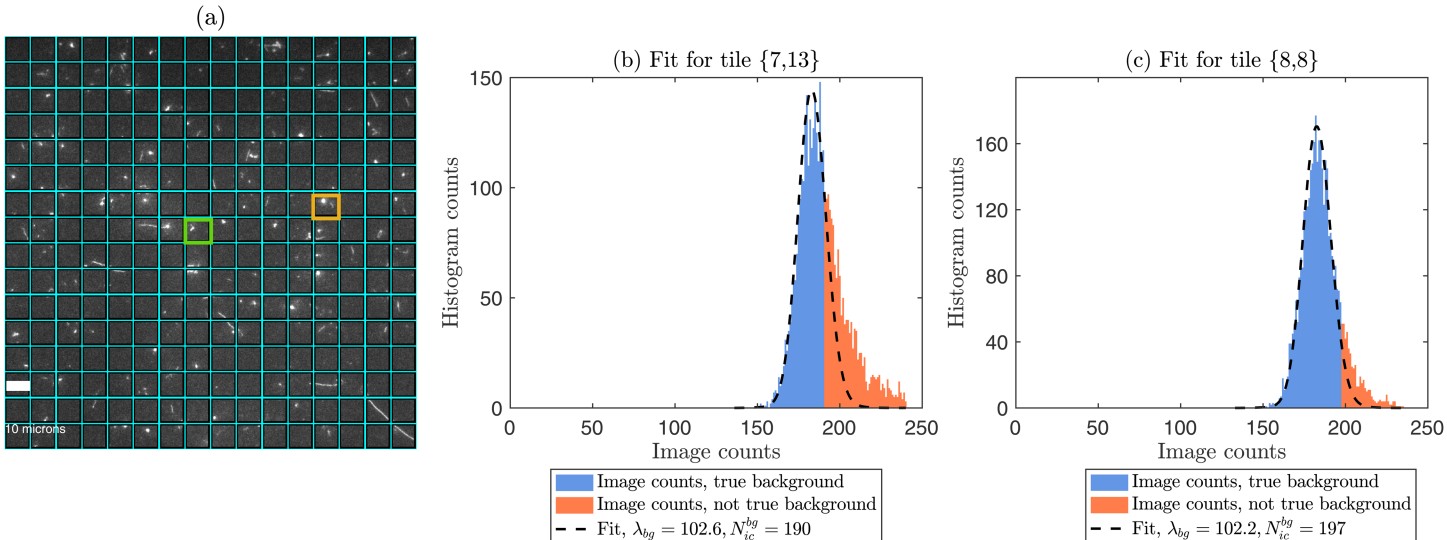

**Fig 3. (a) An sCMOS image of fluorescently labelled single DNA molecules deposited on a glass slide.** The image is acquired using a procedure described in the subsection *Experiments* in Materials and Methods. The image is split into tiles of size 64x64 pixels, where each tile is given a label {row,column}, where in this example row,column = 1,...,16. (b) A histogram of the image counts for a single tile, here tile {7,13} (yellow bordered tile in Fig 3(a)). The blue bars represent pixels regarded as true background, while the orange bars represent the outliers (not true background or signal pixels). The background Poisson parameter is estimated to $\lambda_{bg}$ = 102.6. The image count threshold was estimated to be $N_{ic}^{bg}$ = 190. This threshold separates the blue and orange bars and was determined using a p-value threshold, $p_{GoF}$ = 0.01, for the goodness-of-fit tests. The dashed black curve shows the fitted PMF for the estimated background, extended to the full range of image counts (in our method, we fit a truncated PMF to the blue bars). (c) To show the contrast across the tiles in the image, we estimate $\lambda_{bg}$ = 102.2 for another tile {8,8} (green bordered tile in Fig 3(a)) with $N_{ic}^{bg}$ = 197. Thresholded and segmented versions of the image from panel (a) are found in the Supporting information, S1 Fig.

One of the key advantages of our algorithm is its performance with very low-exposure images (e.g., 1 ms). To demonstrate this, we plotted the sCMOS-PMF fit on the image histogram of the sCMOS camera for three low exposure times: 1 ms, 8 ms, and 20 ms in Fig 4. The average background Poisson parameter for the 1 ms image is $\lambda_{bg}$ = 0.335, indicating that, on average, each pixel records less than 1 photoelectron.

Finally, to further validate the robustness of our sCMOS-PMF fitting procedure, we calculated $\lambda_{bg}$ from DNA on glass images across a wide range of exposure times, from 1 ms to 1000 ms. Ideally, the algorithm should provide a background Poisson parameter that follows a linear relationship with exposure time (10 times longer exposure times results in 10 times more photons hitting the sensor). Fig 5 shows this expected behavior, where $\lambda_{bg}$ indeed display a linear relationship with exposure time.

## Probabilistic image thresholding and image segmentation

With the camera parameters and $\lambda_{bg}$ estimated, we next apply our probabilistic image thresholding and segmentation methods, see "Materials and Methods". Fig 6(a) show an example of a bacterial cell image, acquired using a procedure described in the subsection *Experiments*. Fig 6(b) display a thresholded (binarized) version of this image, where white pixels ideally represent non-background regions (given a significance level set by $p_{binarize}$). From the binarized image we perform image segmentation, where Fig 6(c) displays detected regions (yellow boundaries) using the method described in "Methods and Materials".

We notice that visually our unsupervised thresholding and segmentation procedure works very well for the above example. To further evaluate the robustness and performance of our image processing pipeline, we applied it a few more datasets including bacterial cell images

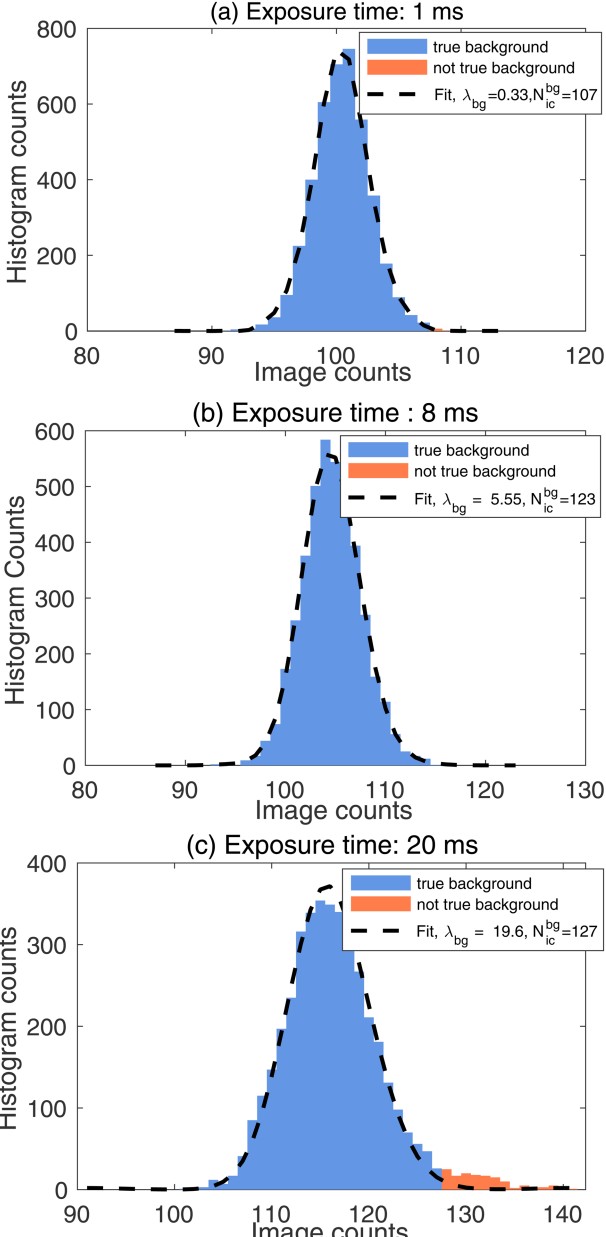

**Fig 4. Performance of the sCMOS-PMF algorithm at low exposure time images: (a) 1 ms, (b) 8 ms, and (c) 20 ms (tile $\{5,7\}$).** The sample being imaged is identical to the one in Fig 3. Black dashed curves represent fitted PMFs. Estimated background Poisson parameters ($\lambda_{bg}$) and threshold counts ($N_{ic}^{bg}$) are given in the figure legends for each case.

acquired at other exposure times and fluorescently stained DNA on glass substrates (Fig 3) (see Supporting information, S2–S3 Figs).

## Comparison of estimates of $\lambda_{\mathbf{bg}}$ for sCMOS and EMCCD for cameras focusing on same FOV

To test the robustness of our algorithm across different fluorescence cameras, we conducted a procedure *Dual camera same FOV experiment* (see the experiments subsection in "Methods"),

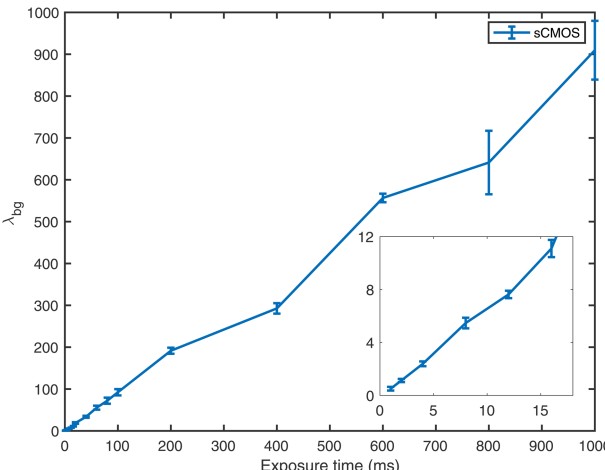

**Fig 5. Relationship of the estimated background Poisson parameter ($\lambda_{bg}$) with the exposure time of the sCMOS camera.** We show the average background Poisson parameter (over all tiles) for images with exposure times (1–1000 ms). The inset shows zoomed version of the image for 1–16 ms. Notice that $\lambda_{bg}$ increases linearly with exposure time, as it should (since the number of collected photons increases linearly with exposure time). The error bars are the standard deviations of $\lambda_{bg}$ across tiles in the image.

where both cameras were mounted on the same microscope and focused on the same field of view (FOV) containing a DNA sample on glass. This approach ensures that, under identical experimental conditions (same light source intensity, and exposure time), both cameras should ideally record the same amount of background photon counts per unit area after offset and noise parameter corrections.

We compared the sCMOS-derived estimates of $\lambda_{bg}$/pixel area with its counterpart from the EMCCD camera as the pixel size for the sCMOS (11 $\mu$m) differs from EMCCD (13 $\mu$m). For the EMCCD setup, we applied the the EMCCD-PIA algorithm by Krog et al. [10]. Prior to analysis, EMCCD calibration was performed following Krog et al.'s [10] protocol with estimated parameters listed in the caption of Fig 7.

We find that the sCMOS algorithm produces an output for $\lambda_{bg}$ per pixel area which aligns closely with the $\lambda_{bg}$ per pixel area from the EMCCD setup, see Fig 7. Minor differences in background counts were observed between the two cameras, potentially due to differences in their quantum efficiencies, *QE*. Additionally, at very low exposure times, EMCCD images exhibit pixel bleeding [30]. This effect and EM gain-amplified spurious charge could cause a mixing of signal and background, potentially leading to an overestimation of $\lambda_{bg}$ for EMCCD cameras at low exposure times, in agreement with the findings in Fig 7 [31,32]. Furthermore, EMCCDs exhibit lower dark current due to their deeper cooling , which reduces thermal noise accumulation in long-exposure images [33]. This lower dark current may contribute to a reduced effective photon count in high-exposure scenarios for EMCCD compared to sCMOS cameras 7 [20,34].

This diligent setup and comparison confirm that, while slight variations exist, our algorithms are robust across different fluorescence camera systems.

## Summary and outlook

We developed a probabilistic image analysis framework for sCMOS cameras, leveraging the statistical properties of photon emission and detection. By exploiting the multiplicative property of the characteristic functions for independent random variables,

**(a) Original Image**

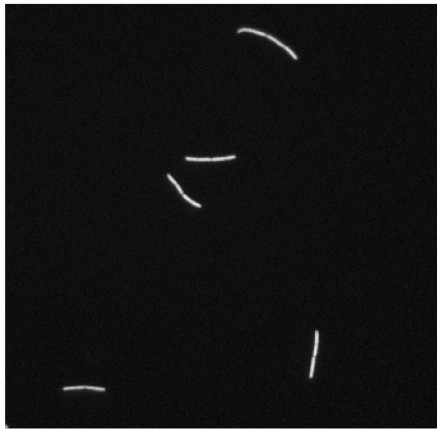

**(b) Thresholded Image**

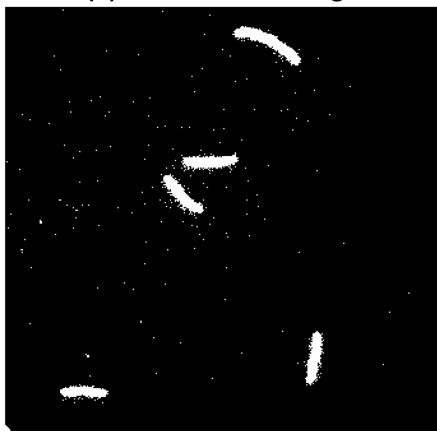

**(c) Segmented Image**

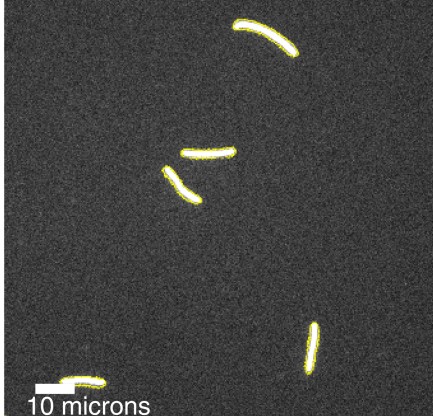

**Fig 6. The photophysical sCMOS image processing pipeline applied to bacteria cells overexpressing GFP.** (a) a 100 ms exposure time image of fluorescently stained cells (balanced gain setting). (b) Binarized image processed by our unsupervised thresholding algorithm with a p-value threshold of $p_{binarize} = 0.01$. Our algorithm is designed so that for this choice of threshold we expect approximately 1% false positives, which from visual inspection may roughly be the case (no ground truth is available here). (c) Output of our segmentation approach with the yellow pixels forms the boundary of the "objects" identified by our unsupervised segmentation method. Example images at lower and higher exposure times are found in the Supporting information, S2–S3 Figs.

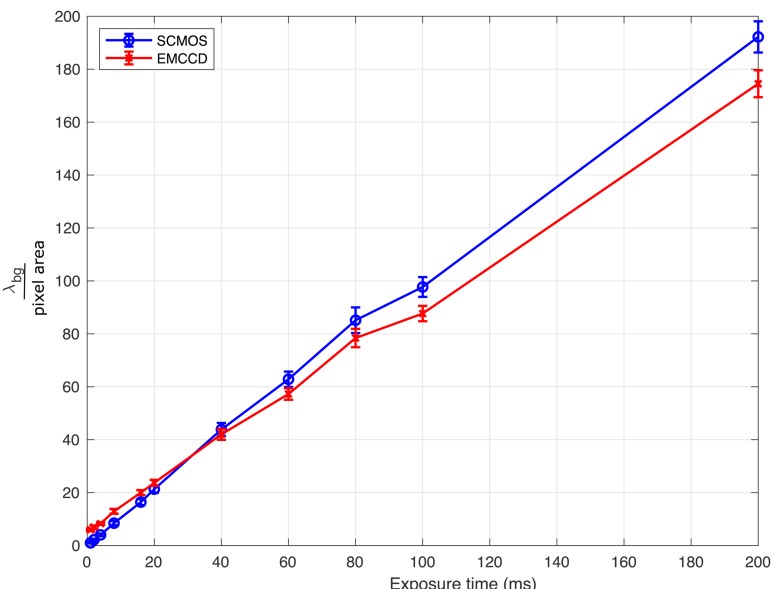

**Fig 7. Comparison of the mean background Poisson parameter per pixel area ($\frac{\lambda_{bg}}{\text{pixel area}}$) between sCMOS and EMCCD cameras.** Images were recorded by sCMOS and EMCCD cameras focusing on same field of view (see *Dual camera same FOV experiment* in experiment subsection). For each camera, $\lambda_{bg}$ was first calculated in individual image tiles; plotted values represent the mean over all tiles, with error bars showing the standard deviation of $\lambda_{bg}$ across tiles. The Poisson parameter is scaled by the pixel area as the pixel size is different for the two cameras (see Experiments section). The calibration parameters for the sCMOS camera are same as calculated in the parameter estimation section while for the EMCCD camera the EM Gain knob on the camera was set at 100. The EMCCD calibration parameters , gain/AD factor ($g/f$), offset ($\Delta$), read noise ($r_0$) are 4.518 , 483.77, and 7.81 respectively, calculated following the procedure discussed for EMCCD cameras in [10].

we derived an expression for the sCMOS-specific probability mass function (sCMOS-PMF) by numerical inversion. This PMF enables estimation of the background Poisson parameter, $\lambda_{bg}$ representing the average number of photoelectrons for background pixels.

Our algorithm excels in low-intensity regimes ($\lambda_{bg} \approx 1$), where traditional thresholding methods often fail due to overlapping signal and noise distributions. To validate the method, we adapted the EMCCD-PIA pipeline for comparative analysis of EMCCD and sCMOS images under equal experimental conditions. The experiment demonstrated agreement in the estimation of $\lambda_{bg}$ ensuring the robustness of the algorithm.

The framework is not restricted to fluorescence imaging; it generalizes to any sCMOS-acquired image where camera noise parameters (readout noise, gain, offset) can be pre-calibrated and proves particularly valuable for limited photon budget experiments, such as imaging photo-sensitive biological specimens or astronomical observations.

We provide public implementations (GUI and non-GUI versions) enabling experimentalists to: a) directly process raw sCMOS images through automated calibration pipelines and robustly estimate background Poisson parameters ($\lambda_{bg}$), for precise signal-background separation and b) generate thresholded and segmented outputs without any user intervention. The software is available at https://github.com/dibyajyoti41.

## Supporting information

**S1 File. See S1–S3 Figs.**
(PDF)

## Acknowledgments

We thank Ann-Brit Schafer from the Division of Chemical Biology, Department of Life Sciences, Chalmers University of Technology for providing the bacterial cells.

## Author contributions

**Conceptualization:** Dibyajyoti Mohanta, Erik Clarkson, Tobias Ambjörnsson.

**Data curation:** Dibyajyoti Mohanta.

**Formal analysis:** Dibyajyoti Mohanta, Erik Clarkson, Tobias Ambjörnsson.

**Funding acquisition:** Fredrik Westerlund, Tobias Ambjörnsson.

**Investigation:** Dibyajyoti Mohanta.

**Methodology:** Dibyajyoti Mohanta, Radhika Nambannor Kunnath.

**Project administration:** Tobias Ambjörnsson.

**Resources:** Radhika Nambannor Kunnath.

**Software:** Dibyajyoti Mohanta, Albertas Dvirnas, Tobias Ambjörnsson.

**Supervision:** Tobias Ambjörnsson.

**Writing – original draft:** Dibyajyoti Mohanta, Tobias Ambjörnsson.

**Writing – review & editing:** Dibyajyoti Mohanta, Radhika Nambannor Kunnath, Erik Clarkson, Albertas Dvirnas, Fredrik Westerlund, Tobias Ambjörnsson.

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
