## [Decision Letter · Decision Letter 0]

29 Aug 2025

PONE-D-25-34479Photophysical image analysis for sCMOS cameras: Noise modelling and estimation of background parameters in fluorescence-microscopy imagesPLOS ONE

Dear Dr. Mohanta,

Thank you for submitting your manuscript to PLOS ONE. After careful consideration, we feel that it has merit but does not fully meet PLOS ONE’s publication criteria as it currently stands. Therefore, we invite you to submit a revised version of the manuscript that addresses the points raised during the review process.

**ACADEMIC EDITOR: **Considering the peer review reports and a careful reading of the manuscript, it appears that the manuscript requires a few minor revisions before it can be accepted for publication.

We look forward to receiving your revised manuscript.

Kind regards,

Hafiz Muhammad Umer Farooqi

Academic Editor

PLOS ONE

Journal Requirements:

1. Please ensure that your manuscript meets PLOS ONE's style requirements, including those for file naming. The PLOS ONE style templates can be found at https://journals.plos.org/plosone/s/file?id=wjVg/PLOSOne_formatting_sample_main_body.pdf and https://journals.plos.org/plosone/s/file?id=ba62/PLOSOne_formatting_sample_title_authors_affiliations.pdf.

 [T.A. acknowledges a research grant 2022-03475 from the Swedish Research Council. F.W and T.A are grateful for funding from the from the Erling-Persson Foundation and the Swedish Childhood Cancer Fund (Barn- cancerfonden), Grant No. MT2022-003. The computations were enabled by resources provided by the National Academic Infrastructure for Supercomputing in Sweden (NAISS) and the Swedish National Infrastructure for Computing (SNIC) at the national Supercomputing Centre at Link\"oping University partially funded by the Swedish Research Council through grant agreements no. No. 2022-06725 and No. 2018- 05973.]

4. Your abstract cannot contain citations. Please only include citations in the body text of the manuscript, and ensure that they remain in ascending numerical order on first mention.

5. Please include the reference section of your manuscript.

Additional Editor Comments:

Reviewer #1:

Reviewer #2:

Reviewers' comments:

Reviewer's Responses to Questions

**Comments to the Author**

1. Is the manuscript technically sound, and do the data support the conclusions?

Reviewer #1: Yes

Reviewer #2: Yes

2. Has the statistical analysis been performed appropriately and rigorously?

Reviewer #1: Yes

Reviewer #2: Yes

3. Have the authors made all data underlying the findings in their manuscript fully available?

Reviewer #1: Yes

Reviewer #2: Yes

4. Is the manuscript presented in an intelligible fashion and written in standard English?

Reviewer #1: Yes

Reviewer #2: Yes

5. Review Comments to the Author

Reviewer #1: This manuscript presents a well-structured and clearly written framework for unsupervised probabilistic image thresholding in sCMOS cameras. The derivation of the sCMOS-specific probability mass function (PMF) is rigorous, and the camera calibration procedures are carefully designed. The experimental results, including comparisons with EMCCD under matched conditions, are convincing and well illustrated.

One minor concern is the potential impact of photobleaching in the dual-camera experiment (Figure 7), as the sCMOS images were acquired before EMCCD. While the authors mention steps to mitigate bleaching, it would be helpful to clarify whether the acquisition order was counterbalanced or to briefly discuss the lower λbg observed for EMCCD at higher exposure times.

Reviewer #2: I appreciated the updated manuscript that included the full reference list and placement of the Figures within the main text. It was very helpful to properly review the manuscript. The authors have a history of publishing manuscripts to improve the quality of fluorescence microscopy images for analysis. This current study on photophysical image analysis (PIA) for sCMOS cameras is a logical extension of their previous work that introduced PIA with unsupervised probabilistic image thresholding for images acquired by EMCCD cameras. The manuscript is well written, and the authors present the information in a clear manner. This work is suitable for publication in the journal of PLoS One with revisions. My major concern is that it would be highly beneficial to include 1-2 more figures (either in the main text or supplemental) to document the application of their technique to a variety of images. It is difficult to appreciate the image processing pipeline and segmentation from the images in Figure 6. Consider adding Figures like Figures 3 and S10 from their 2024 PLoS One publication, or like Figures from references 2 (Nature Methods. 2017) or 3 (Nature Comm 2020) cited in this manuscript.

Below are my recommended minor revisions that need to be addressed.

Minor Comments:

The 2nd and 3rd authors are designated as contributing equally to this work. Was the first author also an equal contributor, or just the 2nd and 3rd author?

Figure 2. The location of (a) and (b) labels should be moved to the upper left-hand corner inside or outside the graph or by the graph title (like Figure 3).

Figure 3. The locations of tiles {7,13} and {8,8} in (a) are unclear. It would be helpful to mark these tiles with different colored outlines to make it easier for the reader to compare the histogram in (b) to tile {7,13} and the histogram in (c) to tile {8,8}.

Figure 4. The location of (a), (b), and (c) labels should be moved to the upper left-hand corner inside or outside the graph or by the graph title (like Figure 3).

There are at least 2 reference that are incorrect (see below). Please carefully review all references for accuracy.

Reference 3 (line 470) is cited as “Nature Communications. 2020;11:238” and should be “Nature Communications. 2020;11:94”

Reference 10 (lines 489-490) is cited as “PLoS ONE. 2024;19(4):e0298321” and should be “PLoS ONE. 2024;19(4): e0300122”

6. PLOS authors have the option to publish the peer review history of their article (what does this mean?). If published, this will include your full peer review and any attached files.

Reviewer #1: No

Reviewer #2: **Yes: **Stephen I. Lentz

---

## [Author Response · Author response to Decision Letter 1]

26 Sep 2025

\section{Reviewer 1}

\noindent{\bf Comment}: This manuscript presents a well-structured and clearly written framework for unsupervised probabilistic image thresholding in sCMOS cameras. The derivation of the sCMOS-specific probability mass function (PMF) is rigorous, and the camera calibration procedures are carefully designed. The experimental results, including comparisons with EMCCD under matched conditions, are convincing and well illustrated.\\

\\

\noindent{\bf Reply}: We thank reviewer for the critical evaluation of the manuscript and generous comments regarding publication of this manuscript in PLOS One. \\

\noindent{\bf Comment}:One minor concern is the potential impact of photobleaching in the dual-camera experiment (Figure 7), as the sCMOS images were acquired before EMCCD. While the authors mention steps to mitigate bleaching, it would be helpful to clarify whether the acquisition order was counterbalanced or to briefly discuss the lower $\lambda_{\text{bg}}$ observed for EMCCD at higher exposure times.\\

\noindent{\bf Reply}: In the revised manuscript (subsection "Experiments"), we clarified the order at which the images were acquired (if an image was captured at one exposure setting first with sCMOS and then EMCCD, the image for the next exposure setting was captured first with EMCCD and then sCMOS). We also elaborated on potential causes for why the estimates of $\lambda_{bg}$/(pixel area) are slightly different for the two camera types (please see the discussion at the end of the Results section). \\

\section{Reviewer 2}

\noindent{\bf Comment}:

I appreciated the updated manuscript that included the full reference list and placement of the Figures within the main text. It was very helpful to properly review the manuscript. The authors have a history of publishing manuscripts to improve the quality of fluorescence microscopy images for analysis. This current study on photophysical image analysis (PIA) for sCMOS cameras is a logical extension of their previous work that introduced PIA with unsupervised probabilistic image thresholding for images acquired by EMCCD cameras. The manuscript is well written, and the authors present the information in a clear manner. This work is suitable for publication in the journal of PLoS One with revisions.\\

\\

\noindent{\bf Reply}:

We thank the reviewer for reading our manuscript and for recommending publication of this manuscript in PLOS One after revisions.\\

\\

\noindent{\bf Comment:}

My major concern is that it would be highly beneficial to include 1-2 more figures (either in the main text or supplemental) to document the application of their technique to a variety of images. It is difficult to appreciate the image processing pipeline and segmentation from the images in Figure 6. Consider adding Figures like Figures 3 and S10 from their 2024 PLoS One publication, or like Figures from references 2 (Nature Methods. 2017) or 3 (Nature Comm 2020) cited in this manuscript.\\

%\noindent{\bf Reply:} We need sCMOS captured image of nuclei (like one in jens plos one paper )\\

\\

\noindent{\bf Reply:} We recorded images of stained bacterial cells fixed on a glass side, and applied our analysis pipeline to these. In the revised manuscript, one of these images (100 ms exposure time) replaced the image of DNA on glass (which is now placed in the Supplementary). Further images (low and high exposure times images) are shown in the Supplementary. We thank the reviewer for this suggestion, which shows the versatility of our method. \\

\\

\noindent{\bf Comment:}

The 2nd and 3rd authors are designated as contributing equally to this work. Was the first author also an equal contributor, or just the 2nd and 3rd author?\\

\noindent{\bf Reply:} The first author is a "sole" first author. it is only the 2nd and 3rd authors which contributed equally. Thus, authors order and equal contributions were correctly noted in the previously submitted version.\\

\noindent{\bf Comment:}

Figure 2. The location of (a) and (b) labels should be moved to the upper left-hand corner inside or outside the graph or by the graph title (like Figure 3).\\

\noindent{\bf Reply:}\\

We have corrected the labels.\\

\\

\noindent{\bf Comment:}

Figure 3. The locations of tiles {7,13} and {8,8} in (a) are unclear. It would be helpful to mark these tiles with different colored outlines to make it easier for the reader to compare the histogram in (b) to tile {7,13} and the histogram in (c) to tile {8,8}.\\

\noindent{\bf Reply:}

We have marked the tiles {7,13} and {8,8} with yellow and green border respectively. We thank the reviewer for this suggestion, which improved clarity.\\

\\

\noindent{\bf Comment:}

Figure 4. The location of (a), (b), and (c) labels should be moved to the upper left-hand corner inside or outside the graph or by the graph title (like Figure 3).\\

\noindent{\bf Reply:}

We have corrected Figure 4. labels.\\

\\

\noindent{\bf Comment:}

There are at least 2 reference that are incorrect (see below). Please carefully review all references for accuracy.

Reference 3 (line 470) is cited as “Nature Communications. 2020;11:238” and should be “Nature Communications. 2020;11:94”

Reference 10 (lines 489-490) is cited as “PLoS ONE. 2024;19(4):e0298321” and should be “PLoS ONE. 2024;19(4): e0300122”\\

\noindent{\bf Reply:}

Thanks for the particular comment. We have rectified the references.

---

## [Decision Letter · Decision Letter 1]

9 Oct 2025

Photophysical image analysis for sCMOS cameras: Noise modelling and estimation of background parameters in fluorescence-microscopy images

PONE-D-25-34479R1

Dear Dr. Mohanta

We’re pleased to inform you that your manuscript has been judged scientifically suitable for publication and will be formally accepted for publication once it meets all outstanding technical requirements.

Kind regards,

Hafiz Muhammad Umer Farooqi

Academic Editor

PLOS ONE

Additional Editor Comments (optional):

After careful review and consideration of the reviewers' conclusive comments, it has been decided that the manuscript meets both the publication and scientific criteria of PLOS ONE and should therefore be accepted for publication.

However, a few minor issues remain regarding image labeling and numbering. These should be corrected during the pre-publication process. As these are minor editorial corrections, they do not require further peer review.

Reviewers' comments:

Reviewer's Responses to Questions

**Comments to the Author**

1. If the authors have adequately addressed your comments raised in a previous round of review and you feel that this manuscript is now acceptable for publication, you may indicate that here to bypass the “Comments to the Author” section, enter your conflict of interest statement in the “Confidential to Editor” section, and submit your "Accept" recommendation.

Reviewer #1: All comments have been addressed

Reviewer #2: All comments have been addressed

2. Is the manuscript technically sound, and do the data support the conclusions?

Reviewer #1: Yes

Reviewer #2: Yes

3. Has the statistical analysis been performed appropriately and rigorously?

Reviewer #1: Yes

Reviewer #2: Yes

4. Have the authors made all data underlying the findings in their manuscript fully available?

Reviewer #1: Yes

Reviewer #2: Yes

5. Is the manuscript presented in an intelligible fashion and written in standard English?

Reviewer #1: Yes

Reviewer #2: Yes

6. Review Comments to the Author

Reviewer #1: (No Response)

Reviewer #2: I appreciated the authors’ effort to address all the comments of both reviewers. Thank you for the updated figures, supplemental figures, and additional references. They are very useful.

Below are a few recommended minor revisions that would be helpful but would not require further review to accept this manuscript for publication.

Minor Comments:

Page 22/65 line 427 (or 54/65 line 427) “(see SI).” should be “(see Supplementary Information, Figs S2-S3).”

Figure 6 Legend. (see page 22/65 or page 57/65). “Example images at lower and higher exposure times are found in the Supplementary.“ should be changed to “Example images at lower and higher exposure times are found in the Supplementary Information, Figs S2-S3.”

It would be helpful to define a, b and c in Supplemental Figure Legends such as these examples below. NOTE: you may need to correct details that are not correct from my copy/paste from Fig 6 legend.

Fig S1. sCMOS pipeline performance on fluorescent DNAs on glass with 100 ms exposure time. (a) a 100 ms exposure time image of fluorescently stained DNA (balanced gain settings) (b) Binarized image processed by our unsupervised thresholding algorithm with a p-value threshold of pbinarize = 0.01. (c) Output of our segmentation approach with the yellow pixels forms the boundary of the ”objects” identified by our unsupervised segmentation method. This figure is a cropped version of Fig. 3 (a) in main text (25 % at the top and 25 % at the bottom were removed for visual clarity).

Fig S2. sCMOS pipeline performance on low exposure time (10 ms) bacteria cell images. Except for the exposure time, all settings were identical to the ones in Fig. 6 in the main text. (a) a 10 ms exposure time image of bacteria cells over expressing GFP (balanced gain settings) (b) Binarized image processed by our unsupervised thresholding algorithm with a p-value threshold of pbinarize = 0.01. (c) Output of our segmentation approach with the yellow pixels forms the boundary of the ”objects” identified by our unsupervised segmentation method.

Fig S3. sCMOS pipeline performance on high exposure time (400 ms) bacteria cell images. Except for the exposure time, all settings were identical to the ones in Fig. 6 in the main text. (a) a 400 ms exposure time image of bacteria cells over expressing GFP (balanced gain settings) (b) Binarized image processed by our unsupervised thresholding algorithm with a p-value threshold of pbinarize = 0.01. (c) Output of our segmentation approach with the yellow pixels forms the boundary of the ”objects” identified by our unsupervised segmentation method.

7. PLOS authors have the option to publish the peer review history of their article (what does this mean?). If published, this will include your full peer review and any attached files.

Reviewer #1: No

Reviewer #2: **Yes: **Stephen I. Lentz

---

## [Editor Report · Acceptance letter]

PONE-D-25-34479R1

PLOS ONE

Dear Dr. Mohanta,

I'm pleased to inform you that your manuscript has been deemed suitable for publication in PLOS ONE. Congratulations! Your manuscript is now being handed over to our production team.

Kind regards,

on behalf of

Dr. Hafiz Muhammad Umer Farooqi

Academic Editor

PLOS ONE